# The influence of common testing floor surfaces on force plate data: Implications for standardisation

**Laura Smith**[ID]*, **Paul Jones**[ID]

Directorate of Psychology and Sport, University of Salford, Salford, United Kingdom

* l.smith@salford.ac.uk

## Abstract

Force plate testing is commonly used to assess athlete performance. However, there is limited research on the effect the surface underneath the force plate has on derived variables. The aim of this study was to investigate whether different surfaces underneath a force plate would elicit differences in derived force plate variables using a mechanical testing device. A device was used rather than human participants to ensure controlled and repeatable impacts. The device was used to assess force reduction, peak force, rate of force development (RFD) and contact time across seven common testing surfaces: vinyl, rubber, Olympic lifting platform, ground (CarpetG) and first floor (CarpetF) carpet, Mondo track and a sprung gymnasium floor (Sprung). Significant differences in force reduction, peak force, RFD, and contact time were found between flooring conditions ($p < 0.05$), with large to extremely large effect sizes. Sprung flooring exhibited the highest force reduction and lowest peak forces, while CarpetF demonstrated the lowest RFD and longest contact time. These findings highlight the flooring surface underneath the force plate during testing significantly influenced derived variables. Practitioners should exert caution and consideration to force plate testing location and advocate standardisation in flooring surface in order to ensure consistent and accurate results.

## Introduction

Force plate testing is commonly utilised within the fields of Biomechanics, Sports Science and Strength and Conditioning due to the wealth of information provided. Practitioners utilise force plate testing to gain insight into an athlete's neuromuscular function, fatigue [1], injury risk [2] and ability to perform within sport specific time constraints [3]. Force plate testing has become more prevalent within professional sport due to the advent of affordable, commercially available force plates [4]. Affordable and commercially available force plate systems have been validated compared to industry gold-standard laboratory-grade force plate systems, such as Hawkin

**Data availability statement:** The data underlying the results presented in the study are available from (Figshare; info@figshare.com; https://figshare.com/articles/dataset/Data_set_-_the_influence_of_common_testing_floor_surfaces_on_force_plate_data_xlsx/28163282).

**Funding:** The author(s) received no specific funding for this work.

**Competing interests:** The authors have declared that no competing interests exist.

Dynamics (Hawkin Dynamics, Westbrook, ME) [4,5], ForceDecks (Vald Performance, Australia) [6] and PASPORT (Pasco Scientific, Roseville, CA) [3]. Similarly, agreement has been reported between commercially available automated software; Hawkin Dynamics, ForceDecks and custom MATLAB analyses [7]. Researchers reported strong agreement between Hawkins Dynamics and ForceDecks however in comparison Sparta Science (Sparta Science, Menlo Park, CA) demonstrated systematic overestimations [8].

Whilst validation and agreement between force plate systems have been investigated, there is limited research into the environment in which the portable force plate testing is administered, i.e., the flooring surface. Previously manufacturers required force plates to be fixed on rigid surfaces [9] with ancillary cabling required for amplifier boxes, connections to data acquisition equipment and power cables, further limiting set-up locations.

The use of commercial portable force plate systems, however, enables users to place force plates on any flat and stable surface, often in some cases without the need for any additional cables and power supply requirements. Industry documentation specifies force plates are recommended on hard, stable, flat surfaces with consistency between testing sessions, similarly that soft surfaces will likely provide inaccurate results [10,11]. Despite this, presumably due to convenience and ability to collect data anywhere, social media posts provide anecdotal evidence that force plate systems are being directly placed on a myriad of flooring surfaces during data collection, such as outdoor running tracks, wooden lifting platforms, rubberised gym floors, sprung gymnasium floors, artificial turf and carpeted floors and in some cases with little standardisation in flooring surface between testing sessions. There exists extensive research into the importance of standardisation of data collection procedures in order to collect consistent data within jump testing, including use of arm swing [12], tucking of the legs [13] and method utilised to calculate jump height [14]. Despite this, there exists no research into the effect of testing surface and a lack of associated standardisation for practitioners.

There currently exists extensive research into the effect of surface on ground reaction forces and derived variables when investigating the *superior* surface-force plate interaction (the surface positioned on top of a force plate), including within walking and running [15,16], jump tasks [17], simulated landing [9] and 3G artificial turf [18]. There is however a sparsity of empirical literature investigating the *inferior* surface-force plate interaction (the surface underneath the force plate). To date there exists one industry-led blog post investigating the effect of floor surface underneath the force plate, in which differences in quiet standing bodyweight were evident between different surfaces [19].

Therefore, to increase the understanding of the role of the surface underneath the force plate it is important to investigate the influence the common testing surfaces available within professional sport have on force plate data. Whilst it would be useful to gain insight utilising athletes and real-world sporting tasks, such as vertical jump tasks, due to the inherent variability between trials [8], a mechanical testing device is an appropriate alternative. Mechanical devices are advocated internationally within

multiple international standards, i.e., EN:14808 [20], in order to assess surface properties within professional sport [21,22]. Mechanical devices are deemed a practical and reproducible testing tool to assess surface behaviour in which values obtained can reproduce ground reaction forces equivalent to an athletes' contact with the ground [23].

The aim of this study was to investigate whether different surfaces underneath a force plate would elicit differences in impact testing, in which it was hypothesised that differences between surfaces would be present. The results of this study will aid informing force plate standardisation for applied practice and research.

## Materials and methods

Ethical approval was not required due to the absence of human participants and the use of a mechanical testing device.

### Flooring conditions

Informal discussions with associates with experience within professional sporting training facilities identified appropriate surfaces. Seven flooring surfaces which are commonly available within professional sporting training facilities were utilised: vinyl overlaying concrete as the reference surface (Vinyl); 10 mm thickness Mondo indoor running tartan track (Mondo, Sportflex, Mondo America Inc., Summit, NJ, USA) (Mondo); 18 mm thickness recycled SBR rubber and EPDM rubber (Achieve, PLAE, GA, USA) (Rubber); 90 mm thickness multi-layer construction solid tongue and groove Red Oak Olympic lifting platform (Power Lift, Jefferson, USA) (Olympic); ground floor 3.1 mm carpet (no underlay) on concrete (CarpetG); first-floor mezzanine level 4.5 mm carpet (no underlay) (CarpetF); traditional sports hall sprung wooden floor (Sprung).

### Mechanical testing device

A mechanical device was utilised in order to adhere to a modified methodology to the European Standard EN:14808 [20] for sports surfaces. The device was originally developed for another study to conduct impact testing which adhered to ASTM F1614 [24] (Procedure A) [25]. The device (Fig 1) consisted of a falling mass of 8.5 kg, diameter of 45 mm and a standardised drop height of 107 mm. The mass was manually lifted and dropped through a shaft onto a Hawkin Dynamics force plate (Hawkin Dynamics, third generation, models 454 and 455, Westbrook, Maine, USA) sampling at 1000 Hz. This system incorporates a dual force plate set-up, however within this study only one force plate per system was utilised. The mass and drop height allowed a peak impact force during the reference surface (Vinyl) of 3190.57 N ± < 1%, which is comparable to or greater than observed peak propulsive vertical ground reaction forces during drop jump and counter-movement jump tasks, respectively [4].

Mechanical impact testing was replicated on each test surface by one single operator. A layer of protective surface was positioned on top of the force plate (Fig 1), this surface was the same throughout the duration of testing, therefore consistent across all flooring conditions.

### Mechanical testing protocol

The force plate was placed directly on top of each test surface and a spirit level used to ensure it was stable and level. The force plate was zeroed in between each trial. In accordance with EN:14808 [20] a reference value was recorded on a vinyl overlaying concrete floor (Vinyl) in which eleven impacts were recorded; during analysis the first impact was discarded and the second to the eleventh impact utilised. For all remaining test surfaces three impacts were recorded; the falling mass was lifted within 5 s of the impact to allow the surface to recover and a time interval of 60 ± 10 s between impacts was adhered to; during analysis the first impact was discarded and the second and third impact utilised [20]. In a modification to EN:14808 [20], three trials were collected for each condition, i.e., three reference trials on concrete and three trials for each flooring condition. To support this sampling strategy, pilot testing was conducted using 30 impacts

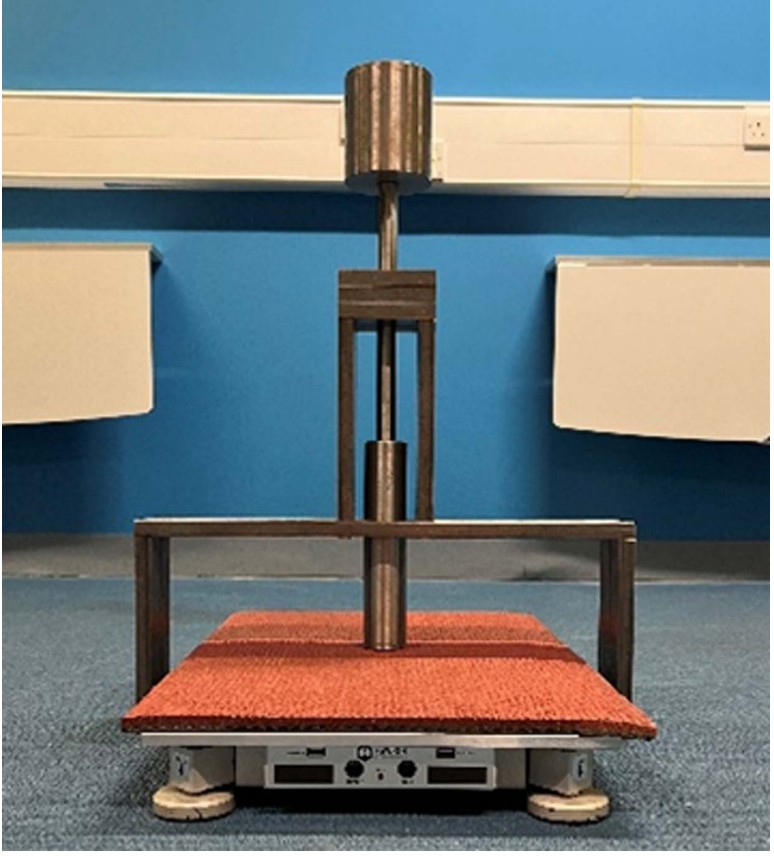

**Fig 1. Example data collection set-up of the mechanical testing device and force plate.**

on the reference surface. Results demonstrated high measurement consistency in peak force with a mean of 3271.27 N, standard deviation of 33.90 N and coefficient of variation of 1.04%. These results, in conjunction with EN:14808 [20] which specifies three impacts per condition are sufficient for impact testing, supported the decision to collect three impacts trials per surface. The mechanical testing protocol was repeated for each flooring condition on two individual Hawkin Dynamics systems to assess agreement between force plate systems in which the device remained in position and the floor plates moved accordingly, to ensure the same location was measured, per condition.

## Data analysis

The unfiltered raw vertical ground reaction force data was exported and analysed using a bespoke Microsoft Excel Spreadsheet (Microsoft Corp., Redmond, WA, USA) to determine force reduction, which is a measure of shock absorption; peak force; rate of force development (RFD); and contact time for each test surface. Peak force was determined as the greatest force (N) during each impact. Force reduction was calculated according to the equation below, in which the peak impact forces obtained during each test surface ($F_t$) were compared to those obtained during Vinyl the reference surface ($F_r$). Force reduction is considered representative of shock absorption of a surface and presented as a percentage [23], the equation is as follows:

$$Force\ reduction = \left(1 - \frac{F_t}{F_r}\right) \times 100 \tag{1}$$

In accordance with Smith and Ditroilo [26] RFD was calculated as the slope of the ground reaction force (GRF) from the onset of force to peak force, the equation is as follows:

$$RFD = \frac{PF - OF}{time_{PF} - time_{OF}}$$

(2)

Where PF is peak force, OF is the onset of force, which was determined when the GRF exceeded a threshold of 25 N and $time_{PF}$ and $time_{OF}$ correspond with the times which PF and OF occurred, respectively. Contact time for each impact was determined as the duration in which the GRF exceeded 25 N, as per manufacturer software settings.

## Statistical analyses

Statistical analysis was conducted using SPSS (version 28, SPSS Inc., Chicago, IL, USA). Fixed and proportional bias between force plates were determined using ordinary least product regression (OLPR) using SPSS (version 28, SPSS Inc., Chicago, IL, USA). Fixed bias was considered present if the 95% confidence interval for the intercept did not include 0 and proportional bias was considered present if the 95% confidence interval for the slope did not include 1. No fixed or proportional bias were present for the measurement variables between the two force plates as the 95% confidence interval for the intercept passed through 0 and the 95% confidence interval for the slope passed through 1, respectively (Table 1); therefore, data from both force plates were pooled for further analyses.

Reliability of each dependant variable for the test surfaces were examined using Intraclass correlation coefficients (ICC) (two-way mixed, single measures, consistency ($ICC_{3,1}$)), standard error of measurement (SEM) and smallest detectable difference (SDD). Magnitude of the ICC was assessed according to Hopkins [27] with the following thresholds: trivial (0.0–0.1), small (0.1–0.3), moderate (0.3–0.5), large (0.5–0.7), very large (0.7–0.9) and nearly perfect (0.9–1.0). SEM and SDD were calculated using the following formula:

$$SEM = SD(pooled) \times \sqrt{(1 - ICC)}$$

(3)

$$SDD = (SEM \times \sqrt{2}) \times 1.96$$

(4)

Normality of distribution was checked for all dependant variables utilising a Shapiro-Wilk test. One-way ANOVA (parametric) or Kruskal-Wallis test (non-parametric) were calculated for all dependent variables, comparing the floor surface conditions. When a one-way ANOVA was conducted, significant differences were located using pairwise comparisons with a Bonferroni correction. When the Kruskal-Wallis test identified significant differences, Mann-Whitney tests were conducted to obtain pairwise comparisons and presented with a Bonferroni correction. Significance was set at $p < 0.05$. Hedges $g$

**Table 1. Descriptive and agreement statistics for selected measurement variables.**

| Variables | HD 454 (Mean±SD) | HD 455 (Mean±SD) | Intercept (95% CI) | Fixed bias | Slope (95% CI) | Proportional bias |
|---|---|---|---|---|---|---|
| Force reduction (%) | 3.22±7.01 | 3.74±6.92 | 0.490 (−0.367 - 1.347) | No | 0.999 (0.801 - 1.196) | No |
| Peak vGRF (N) | 3100±221 | 3091±223 | −37.556 (−444.616 - 368.504) | No | 1.009 (0.880 - 1.138) | No |
| RFD (kNs⁻¹) | 524.84±73.50 | 521.65±68.94 | 29.403 (−271.063 - 329.868) | No | 0.938 (0.405 - 1.471) | No |
| Contact time (ms) | 15.78±1.47 | 15.78±1.53 | −0.621 (−3.673 - 2.430) | No | 1.039 (0.841 - 1.237) | No |

effect sizes were calculated in Microsoft Excel and interpreted as trivial (<0.2), small (0.20–0.59), moderate (0.6–1.19), large (1.20–1.99), very large (2.0–3.99) and extremely large (>4.0) [28].

## Results

Intraclass correlation coefficients demonstrated nearly perfect reliability for force reduction (ICC = 0.986, 95% CI (0.960, 0.998); SEM = 0.89; SDD = 2.47), peak force (ICC = 0.986, 95% CI (0.960, 0.998); SEM = 28.37; SDD = 78.65) and contact time (ICC = 0.929, 95% CI (0.821, 0.998); SEM = 0.42; SDD = 1.17), however RFD demonstrated very large reliability (ICC = 0.773, 95% CI (0.537, 0.955); SEM = 34.41; SDD = 95.38).

Descriptive statistics are presented in Table 2 with visual presentation and pairwise comparisons depicted in Fig 2. In addition, heatmaps in Fig 3 show the Hedge's $g$ values, alongside $p$-values. There were significant differences for force reduction for all flooring conditions ($p < 0.05$). In particular, **Sprung** demonstrated the highest force reduction and lowest peak force compared to other flooring conditions with extremely large effect sizes.

Fewer significant differences were evident for RFD, in which the reference flooring **Vinyl** was significantly different with large to extremely large effect sizes to **Rubber** ($p < 0.01$, $g = 2.21$), **Olympic** ($p = 0.00$, $g = 3.23$), **CarpetF** ($p = 0.00$, $g = 7.61$), and **Sprung** ($p = 0.00$, $g = 5.86$). The reference flooring **Vinyl** contact time was significantly different with large to extremely large effect sizes to **Olympic** ($p = 0.04$, $g = -1.59$), **CarpetG** ($p = 0.00$, $g = -3.17$), **CarpetF** ($p = 0.00$, $g = -9.18$), **Mondo** ($p = 0.04$, $g = -1.59$) and **Sprung** ($p = 0.00$, $g = -2.66$). In particular, **CarpetF** demonstrated the lowest RFD and longest contact time compared to other flooring conditions (Fig 4), with large to extremely large effect sizes.

## Discussion

The aim of the study was to investigate whether the inferior surface-force plate interaction, i.e., surface underneath the force plate would elicit differences in derived variables, using a mechanical testing device. The primary finding of this study was that significant differences in force reduction, peak force, RFD and contact time were present between flooring conditions. The findings of the present study therefore highlight the importance of carefully considering the testing location and the associated flooring surface, when conducting force plate assessments. Furthermore, it is advocated that research utilising commercial force plates explicitly reports the flooring surface used, to enhance reproducibility and ensure accurate interpretation of results.

In accordance with our hypothesis significant differences in peak force between the reference and all other flooring conditions were evident ($p < 0.05$) with large to extremely large effect sizes. These results support previous recommendations highlighting the importance of force plate testing being conducted on rigid surfaces [9–11]. In addition, sports hall flooring (Sprung) exhibited the largest force reduction due to the lowest peak forces, which were significantly different to all other testing surfaces, with extremely large effect sizes. Sports surfaces are designed to deform under stress to increase performance and limit the risk of injury [29] through shock attenuation. The nature of this flooring is to reduce the magnitude and increase the duration of the application of force [29] which is supported by the current study with lower peak forces and longer contact forces compared to the reference flooring (Vinyl). Whilst a sprung flooring type is advocated for sports performance, the shock attenuating properties of this flooring type are not advocated for force plate assessments.

**Table 2. Descriptive data (mean ± SD) for each floor surface condition.**

|  | Vinyl | Rubber | Olympic | CarpetG | CarpetF | Mondo | Sprung |
|---|---|---|---|---|---|---|---|
| Force reduction (%) | – | 1.59 ± 1.23 | 4.85 ± 0.77 | −4.19 ± 0.56 | 4.87 ± 1.09 | −2.83 ± 0.92 | 16.62 ± 0.71 |
| Peak force (N) | 3190.57 ± 29.23 | 3139.83 ± 39.19 | 3035.92 ± 24.68 | 3324.17 ± 18.01 | 3035.25 ± 33.84 | 3281.00 ± 29.23 | 2660.25 ± 22.52 |
| RFD (kNs⁻¹) | 609.79 ± 26.66 | 543.88 ± 41.77 | 517.24 ± 36.02 | 562.16 ± 39.41 | 413.15 ± 18.70 | 568.91 ± 41.88 | 447.58 ± 31.05 |
| Contact time (ms) | 14.53 ± 0.50 | 14.50 ± 0.52 | 15.33 ± 0.49 | 16.00 ± 0.00 | 18.92 ± 0.29 | 15.33 ± 0.49 | 15.83 ± 0.39 |

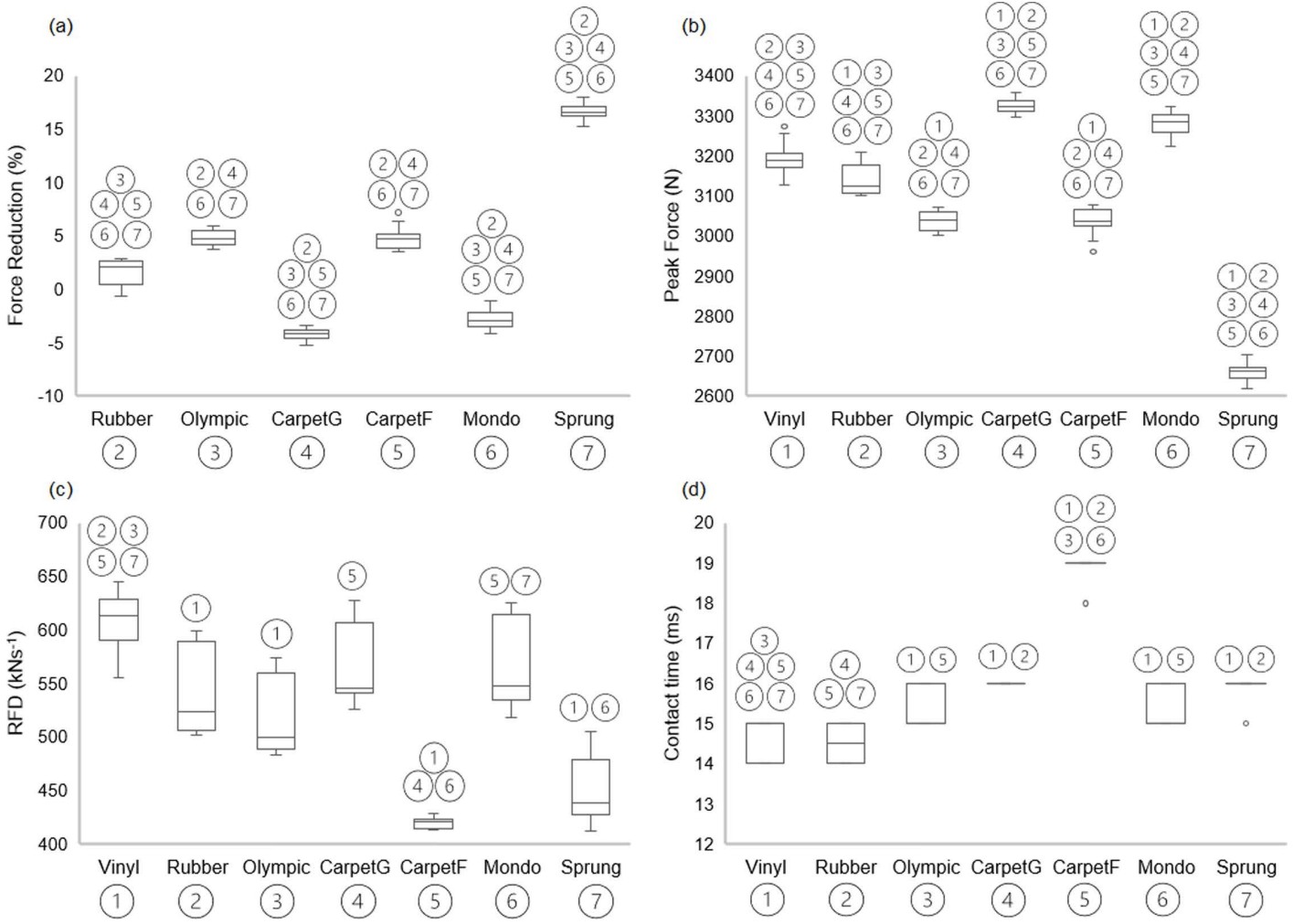

**Fig 2. Boxplots describing each condition and presenting pairwise comparisons between floor surface conditions in (a) force reduction, (b) peak force, (c) RFD and (d) contact time.** Boxplots display the median (centre line), interquartile range (box), minimum, maximum (whiskers) and outliers (dot).

A further example of a shock absorbing surface includes the first-floor mezzanine level carpet (CarpetF), which demonstrated the lowest RFD and longest contact time compared to other flooring conditions, with some significant differences found (Fig 2) with large to extremely large effect sizes (Fig 3). This is likely due to the ability to allow greater surface displacement which would enable force to be distributed over a longer time [18], hence eliciting a reduced RFD due to the longer contact time and lower peak force. This indicates that this flooring type is not suitable for force plate testing and further highlighting the importance of conducting force plate testing on rigid surfaces [9–11].

Contrary to expectations, CarpetG elicited the highest peak force, consequently negative force reduction, with significant differences compared to other flooring conditions (Fig 2) with large to extremely large effect sizes (Fig 3). This anomaly could be two-fold: firstly the reference surface Vinyl differs to the reference surface concrete within EN:14808 [20]; secondly the area of carpet utilised was situated in a high-traffic area of a University building, in which the cushioning properties may have deteriorated. Due to this, and a lack of underlay, the carpet utilised may not be representative of typical carpeted areas available. In retrospect the carpet surface selected within this study should have been subjected

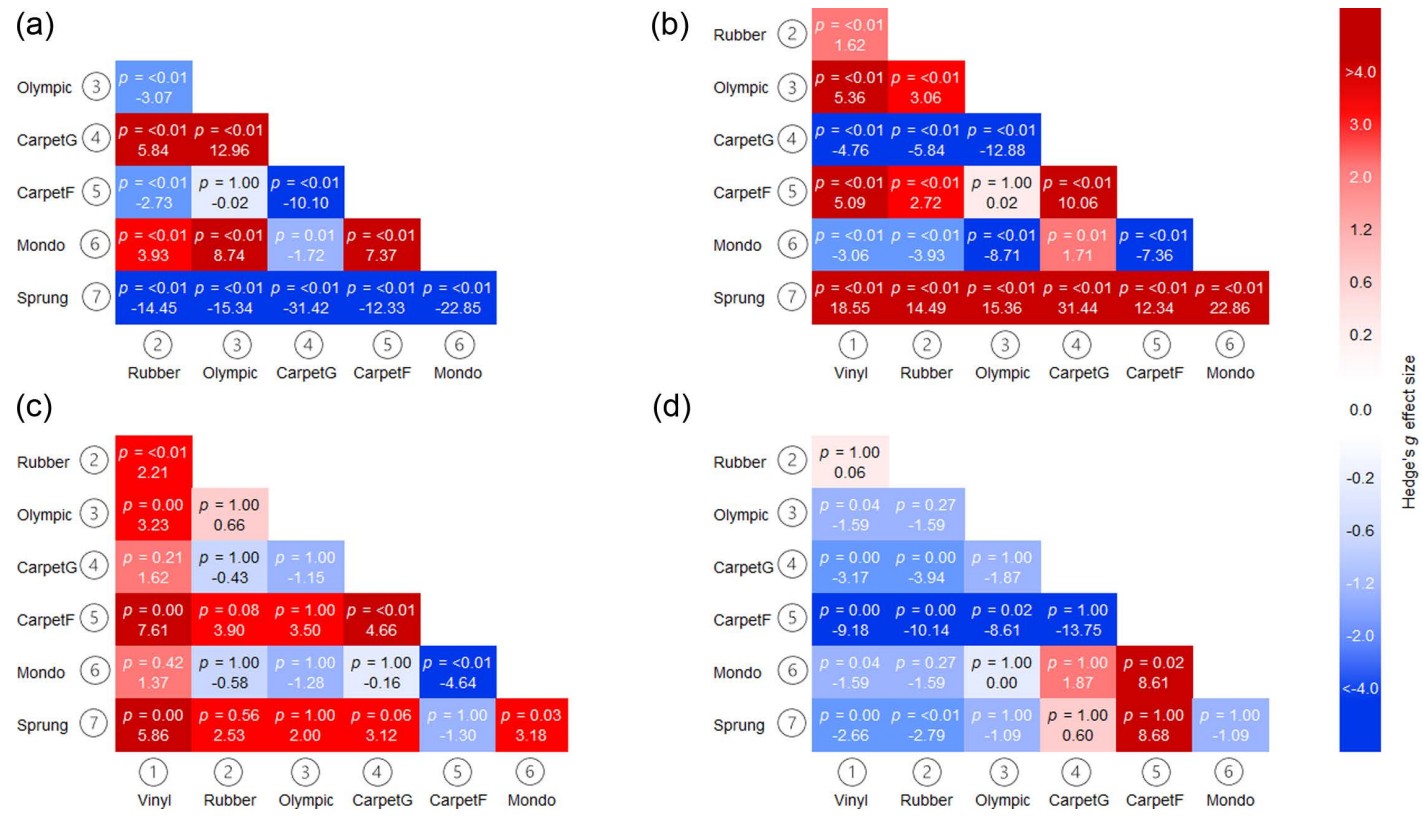

**Fig 3. Heatmaps showing Hedge's g effect sizes and p-values from post hoc pairwise comparisons between surface conditions in (a) force reduction, (b) peak force, (c) RFD and (d) contact time.**

to a preliminary material property assessment. This would have provided insight into carpet properties and whether this specific location was appropriate for inclusion in the current study. Future research should aim to incorporate a material property assessment prior to data collection to provide a more representative carpet sample.

Regarding the use of a mechanical testing device, intraclass correlation coefficients demonstrated nearly perfect reliability for force reduction (ICC = 0.986, 95% CI (0.960, 0.998)), peak force (ICC = 0.986, 95% CI (0.960, 0.998)) and contact time (ICC = 0.929, 95% CI (0.821, 0.998)). In addition, SEM scores were lower than SDD for force reduction (SEM = 0.89; SDD = 2.47), peak force (SEM = 28.37; SDD = 78.65) and contact time (SEM = 0.42; SDD = 1.17), which suggest that the mechanical testing device demonstrates high sensitivity for detecting small and real changes in performance variables. This supports the notion that mechanical testing devices are practical and reproducible testing tools to assess surface behaviour [23]. RFD however demonstrated a lower intraclass correlation coefficient (ICC = 0.773, 95% CI (0.537, 0.955)), higher SEM (34.41) and SDD (95.38): in addition, although significant differences were observed for RFD, there were fewer compared to force reduction and peak force (Fig 2). These factors may reflect the inherent variability of RFD as a performance variable [8].

## Limitations

Although the mechanical testing device was reliable and replicated ground reaction forces equivalent to an athletes contact with the ground [23], a limitation of the present study is that the mechanical testing device eliminated the inherent

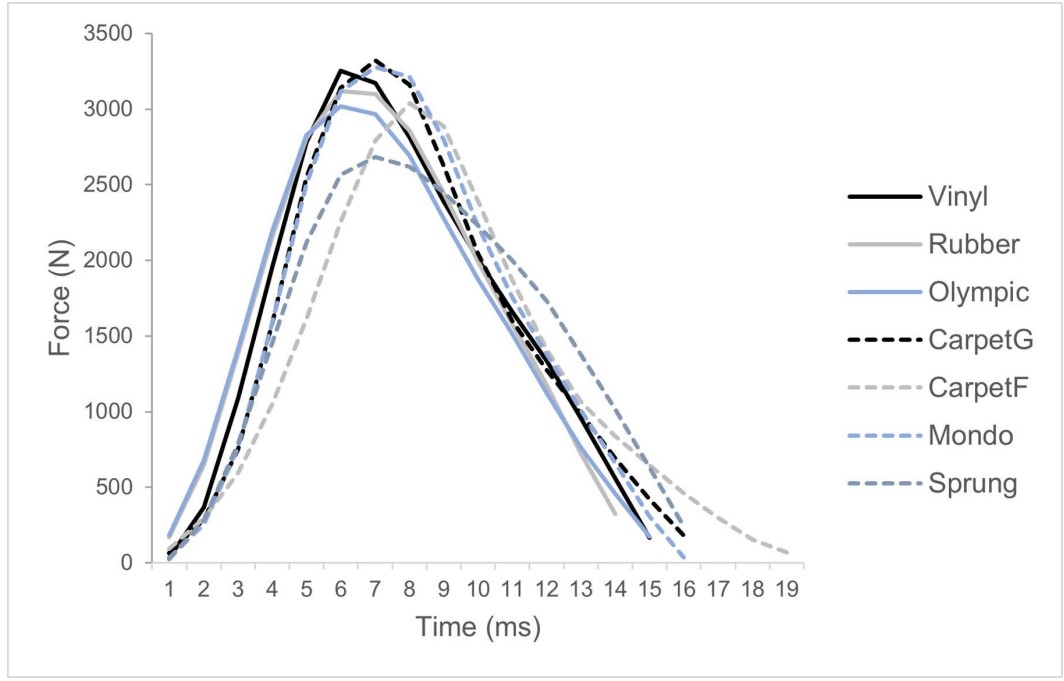

**Fig 4. A representative example of one impact (GRF exceeding 25N) across all flooring conditions.**

variability between trials [8]. In addition to the present study, future research could investigate the effects of different surfaces on performance tasks, such as vertical jump, in real-world athlete testing scenarios.

A further limitation of this study is the surfaces chosen within the current study were selected due to most represented surfaces identified by informal discussions with associates with experience within professional sporting training facilities and anecdotal evidence from social media posts. The selected surfaces however were restrictive in which it would be pertinent to cover further surfaces to represent more diverse sporting environments in which force plate testing may occur, for example, outdoor astroturf, baseball pitch and a professional sprung basketball court.

The findings of this study offer important insights into the influence of the surface underneath the force plate. It is however important to consider that this was conducted utilising a previously custom built mechanical testing device [25] and a specific commercial force plate system (Hawin Dynamics). Whilst the force plate system has been validated compared to industry gold-standard laboratory-grade force plates [4,5], it is important for users to consider the generalisability of these findings to other force plate brands or mechanical testing devices. Therefore, practitioners or researchers using alternative force plate systems should consider conducting impact testing to explore reproducibility of these findings across different systems.

## Conclusion and practical considerations

To the authors' knowledge, there are no existing studies investigating the effect of the inferior surface-force plate interaction, i.e., surface underneath the force plate, during force plate testing. The results of this study have important practical implications for force plate testing standardisation for practitioners within applied practice, in which flooring conditions significantly influenced the derived variables force reduction, peak force, RFD and contact time. Similarly, the results provide important practical applications within research, in which it is advocated that research utilising commercial force plates explicitly reports the flooring surface used. Based on the present study's results caution should be exerted by practitioners

and researchers when considering the location and associated surface underneath a force plate during testing to ensure consistency and accuracy in data collection. It is recommended to utilise a single location and avoid change in surface within or between testing sessions; changes in surface could elicit changes to results which are due to data collection variations and not necessarily changes to performance variables.

Whilst standardising the surface underneath the force plate during testing is ideal for improving data accuracy and integrity, it is not always feasible within the applied and practical working environment. Many practitioners are constrained by time and cost in which testing is often performed under time constraints and with minimal disruption to the athletes. It is pertinent to consider the cost-benefit considerations of standardising flooring surfaces. In some cases, the cost (e.g., time, financial, resource) may outweigh the benefit (e.g., data accuracy). In this scenario, as aforementioned, it is recommended that practitioners document and remain consistent with the surface used during testing. Future research should aim to explore standardisation practices across a broader range of real-world testing environments whilst considering the practical constraints of applied sporting environments.

## Supporting information

**S1 Table. Data set – the influence of common testing floor surfaces on force plate data.**
(XLSX)

## Author contributions

**Conceptualization:** Laura Smith.

**Formal analysis:** Laura Smith.

**Investigation:** Laura Smith.

**Methodology:** Laura Smith.

**Project administration:** Laura Smith.

**Software:** Laura Smith.

**Supervision:** Paul Jones.

**Visualization:** Laura Smith.

**Writing – original draft:** Laura Smith.

**Writing – review & editing:** Laura Smith, Paul Jones.

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
