## [Decision Letter · Decision Letter 0]

23 May 2025

Dear Dr. Smith,

Thank you for submitting your manuscript to PLOS ONE. After careful consideration, we feel that it has merit but does not fully meet PLOS ONE’s publication criteria as it currently stands. Therefore, we invite you to submit a revised version of the manuscript that addresses the points raised during the review process.

We look forward to receiving your revised manuscript.

Kind regards,

Timothy Omara

Academic Editor

PLOS ONE

Journal Requirements:

Additional Editor Comments:

Dear Authors,

The reviewers have now commented on your submission, and recommended reconsideration following minor revision. In addition, I have attached some suggestions to improve the draft in the main MS file.

Reviewers' comments:

Reviewer's Responses to Questions

**Comments to the Author**

1. Is the manuscript technically sound, and do the data support the conclusions?

Reviewer #1: Yes

Reviewer #2: Yes

2. Has the statistical analysis been performed appropriately and rigorously?

Reviewer #1: Yes

Reviewer #2: Yes

3. Have the authors made all data underlying the findings in their manuscript fully available?

Reviewer #1: Yes

Reviewer #2: Yes

4. Is the manuscript presented in an intelligible fashion and written in standard English?

Reviewer #1: Yes

Reviewer #2: Yes

Reviewer #1: The paper is a special one that can become a reference for our field and which corresponds to publication in a journal such as this one, with an extraordinary reputation. The authors managed to synthesize data that can be used for the arrangement of sports structures and not only, the work thus proving to be a well-defined piece of information.

Reviewer #2: The manuscript addresses an important and previously underexplored variable in force plate testing—namely, the influence of inferior surface conditions. The study is original, methodologically rigorous, and has clear relevance for both research and applied practice in sports science and biomechanics. However, a few improvements in structure, clarity, and presentation could enhance the manuscript's impact and readability.

In the abstract, the statement that “there is no research into the effect” might be slightly overstated. It would be more accurate and appropriately cautious to state that “there is limited research,” which acknowledges the broader context while still emphasizing the novelty of the work. Additionally, clarifying early in the abstract that the data were collected using a mechanical testing device, rather than human participants, would improve the reader’s understanding of the experimental model. Including brief details on the number of impacts or trials would also strengthen the impression of methodological robustness.

The introduction is well-motivated and establishes the rationale effectively, but paragraphs two and three are particularly dense and could benefit from being broken into shorter sections to enhance readability. The distinction between superior and inferior surfaces is conceptually important and could be more visually emphasized—perhaps through italics or a parenthetical definition. To further support the practical context described, it might be useful to include at least one additional peer-reviewed source, rather than relying on anecdotal evidence drawn from a blog post, to illustrate the varied surfaces on which force plates are commonly used.

In the Materials and Methods section, while the technical detail is generally strong, it would be helpful to clarify how the specific flooring surfaces were selected—was this based on observational data, informal surveying, or another criterion? Also, the presentation of ICC thresholds is clear and informative, but citing Hopkins (2002) more formally in a complete sentence would enhance the scientific tone. For reproducibility, the exact mass of the mechanical device and the specific model versions of the force plates could be introduced earlier in the section. Furthermore, a brief rationale for the decision to use three impacts per surface, rather than a higher number, would be useful to justify the sampling strategy.

The results are presented thoroughly, and the figures and tables are effective in conveying the key outcomes. However, the narrative describing the pairwise comparisons could be streamlined. Since much of this information is already well visualized in the heatmaps and boxplots, the text might instead focus on summarizing patterns and key implications. In the tables, consider bolding the statistically significant results to help readers scan more efficiently. Figure legends could be slightly expanded so they are interpretable without needing to refer back to the main text. Also, please ensure that color schemes used in heatmaps and plots are accessible for readers with color vision deficiencies, and clarify in the figure captions whether the boxplots reflect mean or median values.

The discussion does a good job of interpreting the findings and placing them in context, but the phrase “Practitioners should exert caution...” is used repeatedly—rewording these instances with some variation would improve readability. The explanation regarding the CarpetG anomaly is a key point of interest and could be expanded further; for example, could a pilot test or a material durability assessment have helped clarify this unexpected result? Including a brief reflection on the cost-benefit considerations of standardizing flooring surfaces in practical field testing environments would also add valuable depth to the applied implications.

The limitations section is appropriate and honest in its scope. However, its visibility could be improved by either including a clearer subtitle or formatting it as a separate, more structured paragraph. It may also help to add a short comment on the generalizability of these findings to other mechanical testing devices or different force plate brands.

The conclusion is clearly written and grounded in the study findings. To strengthen the forward-looking perspective of the manuscript, consider ending with a sentence that encourages future research to explore standardization practices across a broader range of real-world testing environments.

**Do you want your identity to be public for this peer review?** For information about this choice, including consent withdrawal, please see our Privacy Policy

Reviewer #1: **Yes: ** Alin Larion

Reviewer #2: **Yes: ** Valentina Stefanica

---

## [Author Response · Author response to Decision Letter 1]

11 Jun 2025

Thank you for your thorough and constructive feedback. Please see our Response to Reviewers document for our responses.

---

## [Editor Report · Decision Letter 1]

16 June 2025

The influence of common testing floor surfaces on force plate data: implications for standardisation

PONE-D-25-05152R1

Dear Dr. Smith,

We’re pleased to inform you that your manuscript has been judged scientifically suitable for publication and will be formally accepted for publication once it meets all outstanding technical requirements.

Kind regards,

Timothy Omara

Academic Editor

PLOS ONE
---

## [Editor Report · Acceptance letter]

PONE-D-25-05152R1

PLOS ONE

Dear Dr. Smith,

I'm pleased to inform you that your manuscript has been deemed suitable for publication in PLOS ONE. Congratulations! Your manuscript is now being handed over to our production team.

Kind regards,

on behalf of

Dr. Timothy Omara

Academic Editor

PLOS ONE